# FEDCRAP: FEDERATED CRITICAL-REGION-AWARE PERTURBATIONS FOR REFINED PRIVACY-PRESERVING FEDERATED LEARNING

## ABSTRACT

Federated Learning (FL) facilitates collaborative model training across a network of decentralized clients, enabling the development of global models without requiring raw data exchange. This approach preserves data privacy and security by keeping data localized on individual devices, but remains vulnerable to gradient inversion attacks. Existing defense mechanisms rely on global noise injection, which not only causes excessive utility loss or computational overhead but also fails to adequately protect sensitive information requiring additional emphasis. Intensified global perturbations to protect these local sensitive areas can compromise the overall utility of the image. This issue is particularly pronounced in sparse medical imaging data, where critical features are localized in specific regions. To address this challenge, we propose Federated Critical-Region-Aware Perturbations (FedCRAP), a novel defense framework that leverages gradient-guided sparsity patterns. FedCRAP strategically injects noise into task-critical regions identified by high gradient magnitudes, aligning perturbations with the intrinsic sparsity of medical imaging data. By integrating domain-specific sparsity awareness, FedCRAP achieves a favorable balance between privacy preservation and model performance. This provides a finer and more specific noise protection strategy, making it particularly effective. Extensive experiments across various datasets, including sparse medical datasets, demonstrate that FedCRAP preserves model accuracy while significantly reducing privacy leakage risks. It also shows clear superiority over previous state-of-the-art (SoTA) methods for privacy-preserving federated learning.

## 1 INTRODUCTION

In the contemporary era, where artificial intelligence (AI) has achieved significant maturity and widespread application, data has emerged as the most invaluable asset of the information age. AI continues to empower various sectors; however, with its rapid development, researchers have transitioned their focus from merely the volume of data to concerns regarding data privacy and security. In response, numerous countries and regions have enacted legislation to protect user data privacy(Regulation, 2018). Medical data, owing to its sensitive and unique nature, has consistently been at the forefront of privacy protection issues(Act, 1996), leading to challenges in utilizing such data for machine learning training purposes.

Federated Learning (FL), as a distributed machine learning paradigm, implements the design philosophy of "moving models instead of data" (McMahan et al., 2017; Li et al., 2024), enabling collaborative training of global models without sharing local data. This significantly reduces the risk of data privacy leakage and innovatively addresses the core contradiction between data utilization and privacy protection, demonstrating particular value in the medical field.

Although FL's distributed architecture and privacy protection objectives solve the data silo problem, its unique structure and operation mechanism also create opportunities for various attacks. Membership inference (Shokri et al., 2017), attribute inference (Shokri et al., 2017), and Gradient Leakage Attacks(GLAs) (Zhu et al., 2019) represent three typical types of machine learning privacy attacks. Among these, GLAs is particularly prevalent. In FL, each participant keeps their data locally, shar-

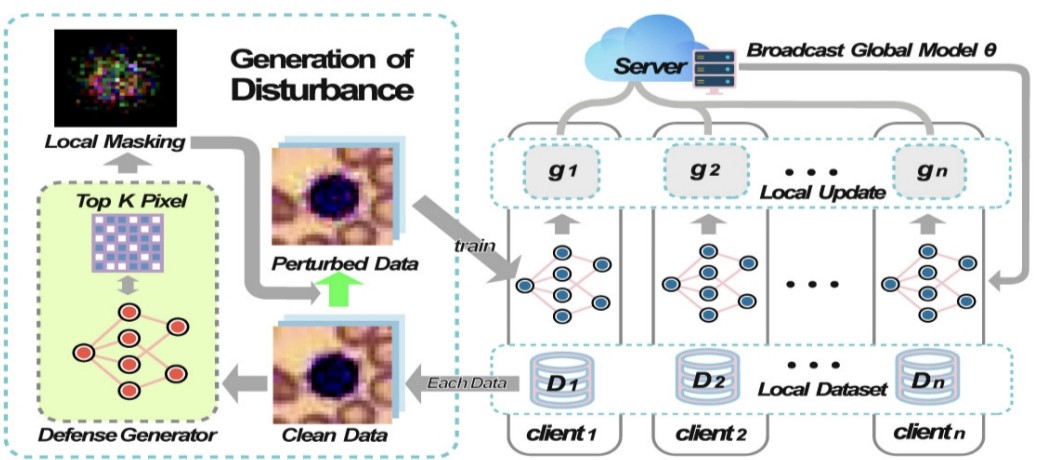

Figure 1: FedCRAP Framework Overview.

ing only model parameters or gradient updates during training. However, gradients contain sensitive information about the data (Lyu et al., 2020). GLAs attempts to recover original data by illegally obtaining shared gradients exchanged between clients and the server, posing risks of sensitive data leakage from FL participants.

Privacy protection remains a core challenge in FL. To mitigate privacy risks, researchers have proposed various methods, primarily including encryption techniques, differential privacy, and its variants such as Centralized DP (CDP) (Geyer et al., 2017) and Local DP (LDP) (Sun et al., 2020). However, these methods still face significant challenges in balancing privacy and utility, where differential privacy becomes a trade-off between noise injection and model performance. Compared to LDP methods that apply global noise in gradient space, FedEM(Xu et al., 2025) innovatively embeds structured perturbations in client data space, achieving a more favorable balance between utility and privacy protection than previous approaches.

These traditional privacy protection methods blindly add noise globally without discrimination, ignoring that information density varies across different regions of an image. Applying equal-intensity noise to non-critical secondary areas not only increases useless information perturbation but may also lead to privacy leaks due to insufficient protection in information-dense key areas. This issue is particularly pronounced in medical data with evident sparsity (Ye & Liu, 2012; Chuang et al., 2007; Huang et al., 2009; Otazo et al., 2015; Davoudi et al., 2019; Fang et al., 2013). To address this challenge, we propose FedCRAP, a novel approach that strategically perturbs high-information-density pixels. Guided by gradient values, FedCRAP sequentially adds differentiated noise across various image regions, tailoring the noise to each image. This targeted perturbation ensures that sensitive areas receive robust privacy protection without over-disturbing non-critical regions, thereby achieving a more favorable balance between utility and privacy. Moreover, considering that medical workflows frequently involve operations such as cropping, FedCRAP's focus on perturbing key image regions ensures that its protective efficacy remains robust despite such modifications.

The primary contributions of this work are as follows:

- **Problem Identification:** We identify a gap in existing FL privacy-preserving methods, which often overlook the unique characteristics of different image regions by applying uniform noise, thereby failing to provide optimal privacy protection, especially in sparse medical images.

- **Methodology:** We propose Federated Critical-Region-Aware Perturbations (FedCRAP), an innovative method that confines perturbations to specific regions, ensuring that models trained on privacy-protected data maintain utility while being resilient against attacks like GLAs.

- **Empirical Validation:** Through comprehensive evaluations across multiple datasets, we demonstrate that FedCRAP achieves a superior balance between utility and privacy in FL systems, offering more refined and effective privacy protection for medical data.

In summary, this paper presents a targeted approach to enhancing privacy protection in federated learning for medical applications, addressing existing shortcomings and paving the way for more secure and efficient collaborative learning in healthcare.

## 2 RELATED WORKS

### 2.1 ATTACK METHODS IN FEDERATED LEARNING

Federated learning enables multiple clients to collaboratively train machine learning models without exposing their raw data. While this decentralized paradigm provides a degree of privacy protection, it does not eliminate all risks. Studies have shown that FL remains vulnerable to privacy leakage through shared model updates. Among various attack vectors, inference attacks, which aim to extract information about the training data were widely studied in early work (Salem et al., 2018; Ganju et al., 2018; Melis et al., 2019; Song & Mittal, 2021). However, as defense mechanisms against such attacks have matured and their practical impact has been reassessed, recent research has shifted its focus to a more severe and direct threat: gradient leakage attacks.

Gradient leakage attacks attempt to reconstruct client-side training data by exploiting the gradients exchanged during training. These attacks are especially concerning in privacy-critical applications such as healthcare and finance. Based on methodology, they can be broadly categorized into two types: gradient analysis attacks and gradient matching attacks.

Gradient analysis attacks exploit the mathematical structure of gradients to analytically recover input data. Aono et al. (2017) first showed that inputs to fully connected layers with bias terms can be exactly recovered from gradients. Zhu & Blaschko (2020) extended this approach to convolutional and bias-free layers, enabling layer-wise reconstruction in deep neural networks. Lu et al. Zhao et al. (2024b) designs a dedicated attack neural network that leverages the reconstructed private data as training samples for secondary learning, enhancing the efficiency and effectiveness of the attack. In the language domain, Gupta et al. (2022) leveraged pre-trained model priors and beam search to reconstruct likely input sequences. Malicious server settings were further explored in (Fowl et al., 2021; 2022), where attackers injected specially designed components or manipulated model weights to enhance gradient leakage.

Gradient matching attacks instead treat data recovery as an optimization problem. Given a set of gradients, the attacker iteratively updates dummy inputs to minimize the discrepancy with the observed gradients. This approach was formalized by Zhu et al.Zhu et al. (2019) as the Deep Leakage from Gradients (DLG) method, Zhao et al.Zhao et al. (2020) proposed iDLG by assuming known labels, which improves convergence and accuracy. Geiping et al. (2020) introduced cosine similarity as a distance metric and demonstrated effective reconstruction of high-resolution images. In NLP, Balunovic et al. (2022) combined gradient matching with language model priors to iteratively refine generated token sequences. More recently, Yue et al. (2023) proposed a framework that integrates low-dimensional feature optimization and image enhancement modules, improving both speed and semantic quality of reconstruction under defense mechanisms.

The above analysis demonstrates that gradient leakage attacks have evolved rapidly in both methodology and scope, encompassing increasingly sophisticated strategies across different model architectures and data modalities. Existing defense mechanisms remain inadequate when confronted with these advanced threats, underscoring the urgent need for more robust and efficient privacy-preserving techniques to safeguard federated learning against evolving privacy risks.

### 2.2 PRIVACY-PRESERVING MECHANISMS IN FEDERATED LEARNING

To address the privacy risks faced by federated learning, a wide range of privacy-preserving techniques have been proposed. The most prominent among them include perturbation-based mechanisms grounded in DP (Reshef & Levy, 2024; Malekmohammadi et al., 2024; Gao et al., 2024;

Wu et al., 2024), and cryptography-based encryption methods (Yan et al., 2024; Kumar et al., 2024; Zhao et al., 2022).

Differential privacy has become one of the most widely adopted privacy-preserving techniques in FL due to its rigorous mathematical definition and verifiable privacy guarantees. First introduced by Dwork et al.Dwork et al. (2006), DP has been extensively applied in FL by injecting carefully designed noise into model updates or gradients to defend against gradient leakage and inference attacks. Depending on the location where the perturbation is added, DP mechanisms in FL can be categorized into LDP(Sun et al., 2020; Liu et al., 2020) and CDP(Miao et al., 2022), where LDP applies noise on the client side, and CDP performs centralized noise addition on the server. Recent works have further advanced DP in FL: Reshef et al.Reshef & Levy (2024) proposed a novel DP mechanism based on stochastic optimization; Malekmohammadi et al.Malekmohammadi et al. (2024) introduced an adaptive noise injection strategy for heterogeneous data scenarios to improve model utility while maintaining privacy; Wu et al.Wu et al. (2024) combined coding techniques with DP to enhance its effectiveness.

In addition to differential privacy, cryptographic approaches have also been explored to ensure privacy in FL, including homomorphic encryption(Yan et al., 2024; Kumar et al., 2024) and secure multi-party computation (SMC)(Zhao et al., 2022). These methods encrypt model parameters or gradients and allow computations to be performed on encrypted data, theoretically ensuring privacy throughout the communication and aggregation process. However, due to their high computational complexity and communication overhead, these techniques remain challenging to deploy at scale in real-world FL systems.

Furthermore, several recent studies have proposed alternative approaches to mitigate privacy risks. For instance, (Zhao et al., 2024a; Maddock et al., 2024) applied data compression techniques to reduce exposure of sensitive information, while (Zhang et al., 2024) explored centralized training schemes to avoid gradient sharing during model updates. While these methods can enhance communication efficiency and reduce data leakage risks, they currently lack systematic theoretical analysis and rigorous privacy guarantees.

### 2.3 PRIVACY-UTILITY TRADE-OFF IN FEDERATED LEARNING

With the continuous development of privacy protection and attack technologies, one of the key challenges faced by Federated Learning (FL) is how to balance privacy protection and model utility. In FL, participants collaborate to optimize a global model by exchanging model parameters or gradient information. Although raw data is not directly shared, the transmitted parameters or gradients may still inadvertently leak sensitive details about the local data. To mitigate privacy risks, additional privacy protection measures are often introduced. However, these measures frequently interfere with the model training process, leading to reduced model performance, slower convergence, or higher resource and communication costs to achieve the same level of accuracy.

Existing research on the trade-off between privacy and utility in FL tends to follow two main directions. On one hand, some studies emphasize model performance but fail to adequately consider the privacy risks involved in the parameter exchange process, which could lead to sensitive data leakage (Fallah et al., 2020; Balakrishnan et al., 2022). On the other hand, other studies focus excessively on privacy protection, adopting stringent privacy mechanisms (such as strong differential privacy parameters or high-intensity encryption), which significantly reduce model accuracy (Wei et al., 2020; Zhu et al., 2021). The existence of these two extreme tendencies indicates that the reasonable balance between privacy protection and model utility has yet to be systematically addressed, becoming a major bottleneck that hinders the widespread application of FL.

Recently, some studies have attempted to analyze the inherent mechanisms of the privacy-utility trade-off from a theoretical perspective. For instance, Zhang et al. Zhang et al. (2022) introduced the "no free lunch" theorem for FL, stating that privacy protection necessarily involves interventions in model updates or parameter information. While such interventions reduce the risk of privacy leakage, they also inevitably lead to a loss in model utility, and vice versa. Additionally, Zhang et al. (2023a) proposed the FedPAC framework, which uses the sample complexity from Probably Approximately Correct (PAC) learning theory to unify the measurement of privacy leakage, utility loss, and training efficiency. This framework transforms the multi-objective optimization problem

into a single-objective optimization problem, simplifying the computational process while providing a detailed analysis of privacy leakage, utility loss, and protection mechanisms.

Despite these efforts, the FL field still lacks a systematic and generalizable theoretical framework and optimization algorithms for the privacy-utility trade-off. Therefore, establishing a scientifically sound analysis framework and achieving personalized balances between privacy protection and model utility for different application scenarios remain critical challenges in FL research.

## 3 FEDERATED CRITICAL-REGION-AWARE PERTURBATIONS FOR PRIVACY PROTECTION

### 3.1 FEDERATED LEARNING

We investigate the privacy preservation problem in federated learning to mitigate sensitive information leakage caused by gradient leakage attacks (GLAs). In a federated learning framework with $K$ clients collaboratively training a global model, each client $i$ holds a local dataset $D_i$. The server maintains global model parameters $\theta \in \mathbb{R}^d$. During the $t$-th communication round, each client computes a local gradient $g_i^{(t)} \triangleq \nabla \mathcal{L}_i(\theta^{(t)})$, where $\mathcal{L}_i(\theta) = \frac{1}{|D_i|} \sum_{(x,y) \in D_i} \ell(f_\theta(x), y)$ represents the empirical loss over $D_i$. The server aggregates these gradients into a global gradient $g^{(t)} = \sum_{i=1}^{K} \alpha_i g_i^{(t)}$, with $\alpha_i = \frac{|D_i|}{\sum_{j=1}^{K} |D_j|}$ weighting the contribution of each client based on its dataset size. This aggregated gradient is used to update the global model parameters via $\theta^{(t+1)} = \theta^{(t)} - \eta g^{(t)}$, where $\eta$ is the learning rate. The process repeats until convergence, aiming to minimize the weighted loss $\mathcal{L}(\theta) = \sum_{i=1}^{K} \alpha_i \mathcal{L}_i(\theta)$.

### 3.2 ATTACK MODEL

The federated learning framework employs gradient-based parameter transmission, where clients and the server exchange gradient vectors instead of raw data or model parameters. Within this architecture, gradients $g_i^{(t)} = \nabla \mathcal{L}_i(\theta^{(t)})$ computed from local datasets $D_i$ are transmitted through potentially insecure channels. A critical vulnerability arises from semi-honest adversaries – entities adhering to protocol specifications while passively intercepting transmitted gradients. These adversaries exploit gradient leakage to launch gradient inversion attacks (Zhu et al., 2019), aiming to reconstruct sensitive input samples $(x, y) \in D_i$ through iterative optimization:

$$\min_{\hat{x},\hat{y}} \left\| \nabla_\theta \ell(f_\theta(\hat{x}), \hat{y}) - g_i^{(t)} \right\|_2^2 \tag{1}$$

This attack paradigm leverages the intrinsic correlation between gradient directions and training data characteristics. Specifically, gradient components corresponding to salient features in $D_i$ exhibit higher magnitudes, enabling adversaries to approximate input patterns through gradient matching.

### 3.3 FEDCRAP

#### 3.3.1 ALGORITHM OVERVIEW

In the domain of Unlearnable Examples, (Sun et al., 2024b) proposed strategically poisoning local regions of medical images to render the data unusable for unauthorized model training—this design effectively blocks unintended learning on sensitive medical data by disrupting task-critical features in targeted areas.

Building on this region-aware insight, we introduce a key creative adaptation: rather than limiting the regional modification strategy to "preventing unauthorized training" (the original goal of unlearnable examples), we transplant this design philosophy into the privacy-preserving scenario of federated learning (FL). More importantly, we reframe the original "poisoning" operation— which aimed to invalidate data for learning—into a dynamic masking mechanism tailored for FL's unique privacy risks. This masked mechanism focuses on protecting critical data regions (instead of disabling

them) and is specifically optimized to defend against Gradient Leakage Attacks (GLAs), directly addressing the core challenge of sensitive data leakage from shared gradients in FL.

To better highlight the innovation of FedCRAP, it is necessary to first clarify the core characteristics of existing representative privacy-preserving methods—including Differential Privacy (DP) variants and FedEM—and their inherent limitations, which FedCRAP is designed to address. DP variants (e.g., DP-Gas, DP-Lap, DP-Clip) are typical perturbation-based methods grounded in differential privacy theory: they inject random noise (e.g., Gaussian, Laplacian) into model gradients or updates in a globally undiscriminating manner—DP-Lap and DP-Gas add noise to gradient values directly, while DP-Clip first clips gradient norms to control sensitivity before adding noise. These methods rely on adjusting privacy budgets $(\epsilon, \delta)$ to balance privacy and utility, but they ignore the heterogeneous information density across different image regions: critical regions (with high task relevance) may receive insufficient noise due to "one-size-fits-all" global perturbation, while non-critical regions are burdened with redundant noise, leading to either privacy leakage or excessive utility loss. FedEM (Xu et al., 2025), a recent data-space perturbation method, improves upon DP by embedding structured perturbations into client-side raw data rather than gradient space, achieving a better utility-privacy balance than traditional DP. However, FedEM still treats the entire image as a uniform entity, failing to distinguish between task-critical and non-critical regions; its structured perturbations are applied globally, which means it still cannot avoid over-perturbing non-essential areas or under-protecting sensitive regions in sparse data (e.g., medical images).

FedCRAP enhances privacy preservation in federated learning by strategically perturbing critical regions of input data identified through gradient sensitivity analysis. Unlike conventional noise injection, FedCRAP introduces a spatially constrained perturbation mechanism guided by a dynamic *mask $M$*, which localizes to the most sensitive and information-dense regions of the image. By precisely adding noise to different local areas of the image multiple times, the sensitive regions of the entire image are well protected. At the same time, excessive noise is not added to unimportant areas. This approach reduces privacy leakage caused by Gradient Leakage Attacks (GLAs) while also maintaining the usability of the data.

The mask is derived from gradient magnitude rankings, ensuring perturbations maximize privacy protection while minimizing utility degradation. The core innovation lies in integrating a bi-level optimization framework with sparsity-aware adversarial training.

### 3.3.2 OBJECTIVE FUNCTION

In the research problem of adding noise to protect data privacy without significantly affecting the utility of the data, we need to consider two optimization objectives:

Model Utility Objective Function Term(Xu et al., 2025):

$$\min_{\boldsymbol{\theta}} \min_{\delta_1, \delta_2, \ldots, \delta_K} \sum_{k=1}^{K} \frac{|\mathcal{D}_k|}{|\mathcal{D}|} \mathbb{E}_{(\boldsymbol{x_k}, \boldsymbol{y_k}) \sim \mathcal{D}_k} \left[ \mathcal{L}\left( f_\theta \left( t \left( \boldsymbol{x_k} + \boldsymbol{\delta_k} \right) \right), \boldsymbol{y_k} \right) \right] \tag{2}$$

Privacy Protection Objective Function Term:

$$\max_{\boldsymbol{\delta_k}} \mathbb{E}_{\hat{\boldsymbol{x}}} \left[ \left\| \hat{\boldsymbol{x}} - \mathcal{A}\left( \tilde{g}\left( \boldsymbol{x_k} + \boldsymbol{\delta_k}, \boldsymbol{\theta} \right) \right) \right\|_p \right] \tag{3}$$

Let $\boldsymbol{\theta}$ denote the global model parameters, $\boldsymbol{\delta_k}$ denotes the perturbation vector for client $k$, $\tilde{g}\left( \boldsymbol{x_k} + \boldsymbol{\delta_k}, \boldsymbol{\theta} \right)$ denotes the gradients in federated learning that have been defended by adding noise, as stolen by DLG attacks, $\mathcal{A}$ denotes Attack model simulating a gradient leakage attack

The Model Utility Objective Function Term aims to minimize the loss of model utility caused by perturbation injection, ensuring that FedCRAP's privacy protection does not undermine the global model's task performance (e.g., diagnostic accuracy for medical images).The Privacy Protection Objective Function Term aims to maximize the difficulty of gradient leakage attacks, ensuring that even if attackers steal the defended gradient $\tilde{g}\left( \boldsymbol{x_k} + \boldsymbol{\delta_k} \right)$, they cannot accurately reconstruct the original sensitive data $x_k$(e.g., patient medical images).

### 3.3.3 Algorithm Details

$M_k$ the binary mask indicating critical regions. We have the following restriction requirements for the added noise:

$$\rho_u^{min} \leq \|M_k \odot \delta_k\|_p \leq \rho_u^{max}, \quad \|M_k\|_0 \leq m. \tag{4}$$

Here, $\rho_u^{min}$ and $\rho_u^{max}$ constrain the allowed norm for perturbation $\delta_k$ , $m$ constrains the sparsity of the mask (i.e., the number of perturbable pixels per perturbation or the percentage of perturbable pixels relative to the entire image.), and $\odot$ denotes element-wise multiplication. The mask $M_k$ is dynamically generated based on gradient magnitudes to prioritize information-dense and privacy-sensitive regions.

Perturbation Update Rule : In each step of the noise generation process, We address the constrained minimization problem in Eq(2) by employing the PGD(Madry et al., 2019) to determine the global noise(which will later be processed into local noise using a mask) added each step:

$$\delta_k^{t+1} = \Pi_{\rho_u^{min} \leq \|M_k \odot \delta_k\|_p \leq \rho_u^{max}} \left( \delta_k^t - \alpha_u \cdot \text{sign}\left( \mathcal{P}\left( G_k, \tau \right)_t \right) \right), \tag{5}$$

where $\tau$ denote the percentage of perturbable pixels relative to the entire image . The direction of noise update in each round is determined by $\mathcal{P}$, ensuring updates only occur in critical regions, which reduces unnecessary distortion . $\mathcal{P}$ will be further introduced below.

Mask Generation : Regarding the generation process of the mask, we employ the method from Sun et al. (2024b) . To better focus on local areas and provide a more refined noise addition strategy, we utilize $M_k$ to locally perturb the image with noise. The process of determining the local regions for each round of noise addition is as follows.

$$\mathcal{P}\left( G_k, \tau \right) = M_k \odot G_k \tag{6}$$

The mask $M_k$ is constructed by thresholding the gradient map $G_k = \nabla_x \mathcal{L}(f_\theta(x_k), y_k)$. For each input $x$, Compute the gradient $G_k$ via backpropagation,Determine the $\tau$-percentile value $g_\tau$ of $G_x$ and generate the mask:

$$M_{(i,j)} = \begin{cases} 1, g_{(i,j)} \geq g_\tau \\ 0, \text{ otherwise} \end{cases} \tag{7}$$

FedCRAP addresses the privacy-utility trade-off in federated learning through spatially aware perturbations. We illustrate the noise addition process for specific data points using the BloodMNIST dataset in Appendix. By integrating gradient sensitivity analysis with sparsity constraints, it achieves stronger privacy guarantees than FedEM while maintaining competitive model performance.

Table 1: Comparison of different methods.

| DATASET | METHOD | Val ACC(U,↑) | Test ACC(U,↑) | Test MSE(P,↑) | Feature MSE(P,↑) | PSNR(P,↓) | SSIM(P,↓) |
|---------|--------|--------------|---------------|---------------|------------------|-----------|-----------|
| | Clean | 0.8560 | 0.8523 | 0.8064 | 0.8206 | 10.5880 | 0.2453 |
| | DP-Gas | 0.6653 | 0.6667 | 0.1919 | 0.0022 | 16.7570 | 0.6170 |
| | DP-Lap | 0.6284 | 0.6297 | 0.1666 | 0.0069 | 17.2290 | 0.5922 |
| FMNIST | DP-Clip | 0.8429 | 0.8365 | 0.5111 | 1.6929 | 14.0510 | 0.3667 |
| | FedEM | 0.8357 | 0.8354 | 0.8213 | 1.7213 | 10.0320 | 0.1454 |
| | **FedCRAP(Ours)** | **0.8634** | **0.8470** | **1.2212** | **6.1792** | **8.6519** | **0.1317** |
| | Clean | 0.9819 | 0.9768 | 0.9324 | 2.0676 | 11.3870 | 0.1573 |
| | DP-Gas | 0.7765 | 0.7813 | 0.6284 | 0.0284 | 13.4210 | 0.4703 |
| | DP-Lap | 0.6550 | 0.6483 | 0.2418 | 0.0120 | 9.7231 | 0.6536 |
| MNIST | DP-Clip | 0.9523 | 0.9502 | **1.2227** | 2.2886 | **9.0826** | 0.1163 |
| | FedEM | 0.9711 | 0.9671 | 1.1800 | **7.7600** | 9.8471 | 0.1131 |
| | **FedCRAP(Ours)** | **0.9768** | **0.9732** | 1.2162 | 6.4864 | 10.2659 | **0.1088** |

## 4 Experiments and Results

### 4.1 FL settings

To evaluate privacy-preserving mechanisms in federated learning (FL), we conduct experiments on three benchmark datasets: MNIST(Deng, 2012)and FashionMNIST(Xiao et al., 2017). These

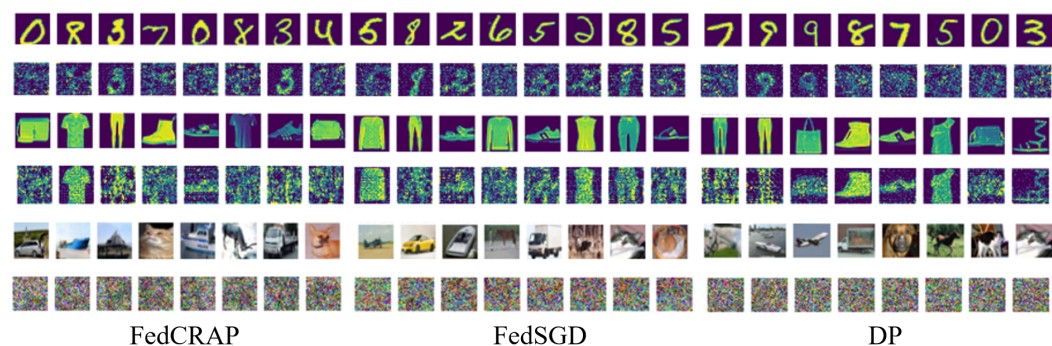

FedCRAP                        FedSGD                          DP

Figure 2: Visualization of the Results of Different Privacy Protection Algorithms Against DLG Attacks

datasets span varying complexities and modalities: MNIST and FashionMNIST comprise 60,000 training and 10,000 testing grayscale images across 10 classes. To emulate real-world FL scenarios, each dataset is partitioned into non-overlapping subsets distributed across 4 clients under an IID split, with an 80-10-10 ratio for training, validation, and testing to ensure reproducibility.

We adopt ResNet-18(He et al., 2016) as the base model, balancing computational efficiency and classification performance. The FL framework follows the FedSGD protocol(McMahan et al., 2017), where clients execute local stochastic gradient descent (SGD) updates with a learning rate of 0.01, momentum of 0.9, and no weight decay to isolate regularization effects. Each client trains for 1 local epoch per communication round using a batch size of 8, while global aggregation operates over 50 epochs to ensure convergence. Early stopping halts training if validation accuracy plateaus for 40 consecutive rounds, mitigating overfitting risks.

## 4.2 EVALUATION METRICS

To evaluate the impact of privacy-preserving methods on dataset utility and investigate the trade-off between privacy protection and data usability(Zhang et al., 2023b), we analyze the performance degradation caused by noise injection strategies. The processed datasets under various privacy mechanisms are partitioned into training and validation sets. The test accuracy (Test-ACC) and validation accuracy (Val-ACC) are employed as primary metrics to quantify utility preservation, where the accuracy drop before and after noise perturbation reflects the extent of utility loss. For privacy assessment, we leverage the DLG attack to reconstruct sensitive images from gradients and calculate four metrics—Test MSE, Feature MSE, SSIM, and PSNR—computed against the original data. These four metrics are used to evaluate the privacy-preserving performance of different privacy protection methods and the details of them can be seen in the Appendix. By integrating accuracy degradation analysis with privacy leakage quantification, we can comprehensively and integrally evaluate the strengths and weaknesses of different privacy protection methods.We demonstrate the relative superiority of our method in the performance-utility-privacy trilemma(Sun et al., 2024a).

## 4.3 EFFECTIVENESS ANALYSIS

Our method achieves enhanced privacy protection through iterative localized noise injection, ensuring data confidentiality after sufficient perturbation steps. To validate the superiority of FedCRAP, we conducted comparative experiments against multiple baseline approaches across three benchmark datasets in federated learning. During training, we launched DLG attacks and evaluated privacy preservation by computing Test MSE, Feature MSE, PSNR, and SSIM between reconstructed and original (attacked) images. Higher Test MSE/Feature MSE and lower PSNR/SSIM values indicate stronger privacy protection. Figure 5 in Appendix illustrates the impact of different noise addition methods on the accuracy of the original data. As can be seen from Figure 5, FedCRAP is not inferior to, and even outperforms other noise addition methods in ensuring that the accuracy is not significantly affected. Under this premise, as shown in Table 1, FedCRAP outperforms its com-

Table 2: Results of Different Algorithms on the PneumoniaMNIST and BloodMNIST

| DATASET | METHOD | VAL ACC(U,↑) | TEST ACC(U,↑) | TEST MSE(P,↑) | FEAT MSE(P,↑) | PSNR(P,↓) | SSIM(P,↓) |
|---------|--------|---------|---------|---------|---------|---------|---------|
| BloodMNIST | FedSGD | 0.8551 | 0.8593 | 2.4234 | 17.7843 | 11.0750 | 0.04133 |
| | DP-Gas | 0.8563 | 0.8599 | 1.6195 | 4.1459 | 12.7538 | 0.0659 |
| | DP-Lap | 0.8580 | 0.8649 | 1.7577 | 4.6171 | 12.5148 | 0.0717 |
| | FedEM | 0.8516 | 0.8532 | 1.6886 | 4.3815 | 12.6343 | 0.0688 |
| | **FedCRAP(Ours)** | 0.8680 | 0.8678 | 1.9378 | 11.8034 | 11.9499 | 0.0432 |
| PneumoniaMNIST | FedSGD | 0.8740 | 0.8782 | 1.7823 | 3.0114 | 13.1104 | 0.0600 |
| | DP-Gas | 0.8511 | 0.8542 | 1.7209 | 1.6266 | 13.3475 | 0.0866 |
| | DP-Lap | 0.8588 | 0.8590 | 1.7262 | 5.1529 | 13.3087 | 0.0868 |
| | FedEM | 0.8779 | 0.8798 | 1.8060 | 3.1425 | 13.0293 | 0.0600 |
| | **FedCRAP(Ours)** | 0.8721 | 0.8750 | 1.8606 | 3.3025 | 13.1915 | 0.0852 |

petitors in most metrics across all datasets, particularly excelling in Feature MSE. As shown in Table 1, FedCRAP performs similarly to the clean method (Clean) in terms of accuracy, but significantly outperforms it in privacy protection.FedCRAP maintains minimal accuracy degradation on MNIST (0.4%), while exhibiting moderate drops on FashionMNIST (1.5%). These results demonstrate that our noise-adding method provides significant privacy protection for the original data.Although Fed-CRAP introduces two core extra steps: dynamic mask generation (based on gradient magnitude ranking and thresholding) and localized perturbation optimization (via a small number of PGD iterations) ,This minor additional overhead is fully justifiable given FedCRAP's advantages.

To further evaluate the robustness and generalizability of FedCRAP, we extended our comparative study to two additional medical imaging datasets: BloodMNIST and PneumoniaMNIST.Across both datasets in Table 2, FedCRAP exhibits a consistent ability to safeguard sensitive medical image features while preserving competitive classification accuracy. On BloodMNIST, it clearly outperforms all baselines in both utility and privacy protection; on PneumoniaMNIST, it delivers a balanced performance that remains highly competitive despite the inherently challenging nature of the dataset and its pre-processing constraints. These findings reaffirm FedCRAP's suitability for privacy-critical medical imaging applications, where both diagnostic accuracy and protection against gradient inversion attacks are paramount.

## 5 CONCLUSION

In this work, we propose the novel method FedCRAP to address the important aspects that previous federated learning privacy-preserving methods have long overlooked. FedCRAP is based on an obvious intuition: if focusing on too many areas at once in a task makes it difficult to optimize every detail, then we only concentrate on the more important parts of the overall image during each noise addition. By focusing on local areas, it is easier to achieve the best results locally. Through multiple rounds of focused processing on different local areas, an optimal noise addition strategy that is refined globally can be achieved.A large number of experiments have shown that, due to the characteristic of FedCRAP processing image pixel subsets in stages, it effectively protects the data participating in federated learning from the risk of data privacy theft, thereby providing significant privacy protection for federated learning systems.

Future work may focus more on exploring more advanced noise addition strategies, as well as integrating the idea of multiple selective processing of important local areas with more diverse federated learning methods. As large models and big data continue to grow rapidly in the future, the demand for privacy and security will become increasingly urgent and stringent. Federated learning emerged to meet the needs of the times, and numerous current studies are continuously strengthening and perfecting the privacy protection mechanisms of federated learning. It is hoped that this paper can contribute, even if only in a small way, to the vast body of research in this field.

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

# A APPENDIX

## A.1 METRICS FOR DLG ATTACKS

The attack efficacy is quantified using four complementary metrics:

**Test mean squared error (TEST-MSE):**

$$\frac{1}{n} \sum_{i=1}^{n} \|x_i - \hat{x}_i\|^2 \tag{1}$$

where $\hat{x}_i$ denotes reconstructed samples, $n$ is the number of pixels. This metric directly measures the pixel-wise accuracy of reconstructed images but may fail to capture perceptual quality . Lower values indicate better reconstruction. The TEST-MSE quantifies pixel-level discrepancies between reconstructed data $\hat{x}$ and original data $x$.

**Feature-level MSE (FEA-MSE):**

$$\frac{1}{d} \sum_{j=1}^{d} \|\phi(x)_j - \phi(\hat{x})_j\|^2 \tag{2}$$

where $\phi(\cdot)$ represents deep features. FEA-MSE uses the pre-trained model $\phi(\cdot)$ to evaluate semantic similarity in feature space. It reflects semantic fidelity by comparing high-level features, addressing limitations of pixel-level metrics. Widely adopted for evaluating privacy leakage in federated learning.

**Structural Similarity Index Measure (SSIM) :** SSIM assesses perceptual quality by comparing luminance ($\mu$), contrast ($\sigma$), and structure ($\sigma_{xy}$):

$$\text{SSIM}(x, \hat{x}) = \frac{(2\mu_x \mu_{\hat{x}} + C_1)(2\sigma_{x\hat{x}} + C_2)}{(\mu_x^2 + \mu_{\hat{x}}^2 + C_1)(\sigma_x^2 + \sigma_{\hat{x}}^2 + C_2)}, \tag{3}$$

where $C_1, C_2$ stabilize division. Values range in $[-1, 1]$, with higher values indicating better structural preservation.

**Peak Signal-to-Noise Ratio (PSNR):** PSNR measures signal fidelity using the maximum pixel value ($L$, typically 255) and TEST-MSE:

$$\text{PSNR} = 10 \cdot \lg\left(\frac{L^2}{\text{TEST-MSE}}\right). \tag{4}$$

Higher values denote lower noise and better signal preservation.

These metrics collectively measure the effectiveness of privacy preservation by evaluating reconstruction fidelity and perceptual similarity.

## A.2 DIFFERENT τ SELECTION

In the previous experiments, we set the value of $\tau$ to 10% by default, and the noise norm radius to 8/255. How should we determine the value of $\tau$ when using FedCRAP in practical applications, so as to maximize its privacy protection capability without affecting the accuracy ? In this experiment, to further investigate the impact of varying $\tau$ values on the performance of FedCRAP, we conducted tests on the FashionMNIST dataset under multiple different training rounds for DLG attacks. By plotting the various metrics exhibited by FedCRAP under different values of $\tau$ into a line chart as shown in Figure 3, we can clearly observe the following phenomenon: the results demonstrate that $\tau = 30\%$ achieves optimal privacy protection on FashionMNIST with ResNet-18, while maintaining data utility (Val ACC and Test ACC) without significant degradation. Remarkably, the slight robustness improvement observed may stem from the reduced noise magnitude introduced by the algorithm. Notably, when $\tau = 100\%$, the method degenerates into a traditional global noise-based privacy-preserving algorithm. Smaller values of $\tau$ can still achieve good protection effects after multiple rounds of noise addition. However, further increasing $\tau$ does not significantly enhance the protection performance. This is precisely the reason why traditional methods fall behind and need improvement: to protect privacy, too much redundant noise is added to the original data.

This experiment demonstrates that the optimal value of $\tau$ is neither the largest nor the smallest, but rather depends on the specific characteristics of the dataset. A reasonable selection of $\tau$ based on these characteristics is essential for achieving the best performance. We plan to further investigate how to determine the optimal $\tau$ value more efficiently for different datasets in future work to achieve the most desirable outcomes.

Given the varying characteristics of images across different datasets, selecting a universal optimal $\tau$ value for all datasets is challenging. We found that setting $\tau$ at 10% effectively preserves privacy without degrading image quality. Thus, for simplicity, we consistently set $\tau$ to 10% in our subsequent experiments.

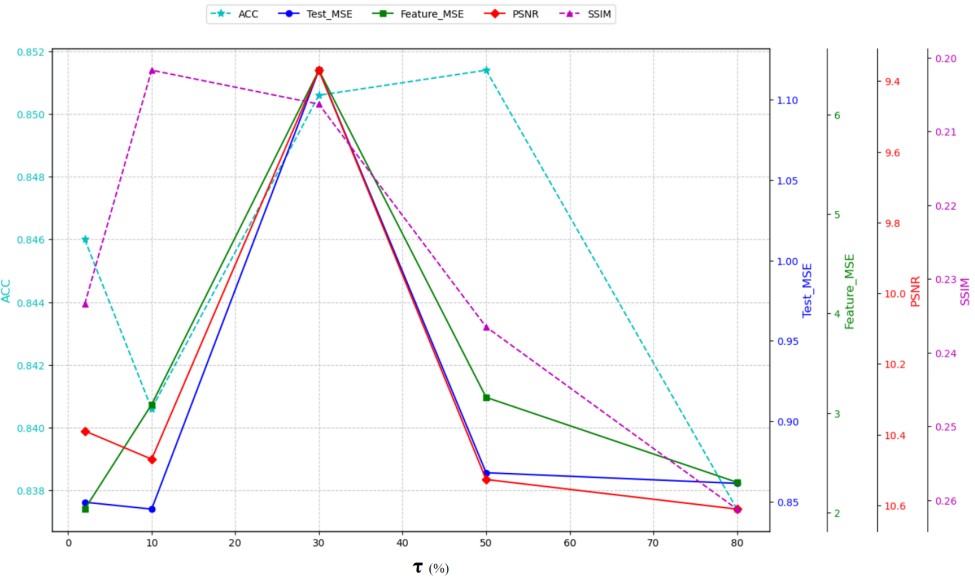

Figure 3: Trend of Result Metrics Changes with Different $\tau$ Selection

## A.3 VISUALIZATION OF THE NOISE ADDITION PROCESS

We illustrate the noise addition process for specific data points using the BloodMNIST dataset in Figure 4.

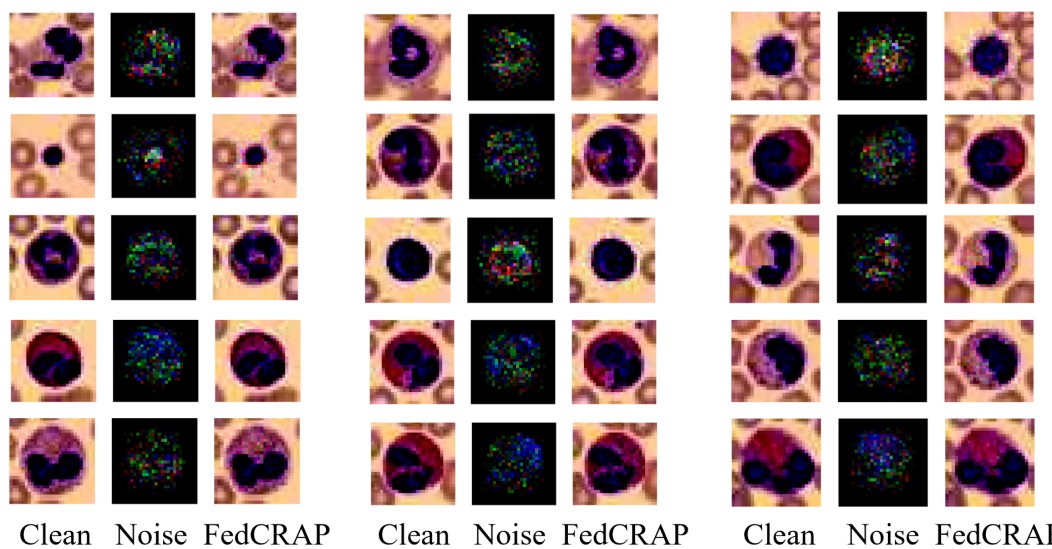

Clean  Noise  FedCRAP  Clean  Noise  FedCRAP  Clean  Noise  FedCRAP

Figure 4: Visualization of the Noise Addition Process.(In each group of images, the leftmost image is the original, the middle image visualizes the processed noise, and the rightmost image is the result after adding one round of noise using FedCRAP)

Table 3: Comparison of different radius on medical datasets

| DATASET | METHOD | VAL ACC(U,↑) | TEST ACC(U,↑) | TEST MSE(P,↑) | FEAT MSE(P,↑) | PSNR(P,↓) | SSIM(P,↓) |
|---|---|---|---|---|---|---|---|
| OCTMNIST | FedEM(1024) | 0.8312 | 0.8452 | 1.7329 | 13.3718 | 11.9857 | 0.0337 |
| | FedEM(2048) | 0.8280 | 0.8431 | 1.5912 | 16.3594 | 12.4864 | 0.0223 |
| | FedEM(4096) | 0.8153 | 0.8292 | 1.4340 | 18.3011 | 12.6001 | 0.0289 |
| | FedCRAP(1024) | 0.8322 | 0.8486 | 2.5352 | 13.2344 | 10.5113 | 0.0257 |
| | FedCRAP(2048) | 0.8154 | 0.8267 | 1.3496 | 23.7580 | 12.8963 | 0.0348 |
| | FedCRAP(4096) | 0.7898 | 0.8004 | 1.9561 | 12.5274 | 11.3061 | 0.0216 |
| BreastMNIST | FedEM(1024) | 0.8086 | 0.7940 | 5.2473 | 0.0564 | 6.6556 | 0.0295 |
| | FedEM(2048) | 0.8112 | 0.7922 | 5.2896 | 0.0435 | 6.5834 | 0.0189 |
| | FedEM(4096) | 0.8138 | 0.7940 | 5.4801 | 0.0406 | 6.3958 | 0.0154 |
| | FedCRAP(1024) | 0.7595 | 0.7336 | 5.8732 | 0.3797 | 6.9222 | 0.0487 |
| | FedCRAP(2048) | 0.7474 | 0.7284 | 6.1988 | 0.4390 | 6.6625 | 0.0424 |
| | FedCRAP(4096) | 0.7534 | 0.7293 | 7.3373 | 0.2173 | 5.9748 | 0.0415 |
| BloodMNIST | FedEM(1024) | 0.6454 | 0.6471 | 1.9938 | 4.0800 | 11.9450 | 0.0562 |
| | FedEM(2048) | 0.4083 | 0.4094 | 2.2248 | 1.9910 | 11.3700 | 0.0307 |
| | FedEM(4096) | 0.3102 | 0.3175 | 3.7957 | 0.0134 | 8.9547 | 0.0146 |
| | FedCRAP(1024) | 0.6466 | 0.6477 | 2.0203 | 3.5262 | 11.7058 | 0.0327 |
| | FedCRAP(2048) | 0.3773 | 0.3722 | 2.2617 | 2.5242 | 11.2494 | 0.0242 |
| | FedCRAP(4096) | 0.3756 | 0.3868 | 3.8523 | 0.0099 | 8.8959 | 0.0145 |
| PneumoniaMNIST | FedEM(1024) | 0.8511 | 0.8446 | 2.4025 | 1.6415 | 11.7792 | 0.0262 |
| | FedEM(2048) | 0.8340 | 0.8285 | 2.3268 | 0.8603 | 11.8668 | 0.0254 |
| | FedEM(4096) | 0.8321 | 0.8317 | 3.8888 | 0.0788 | 9.6960 | 0.0147 |
| | FedCRAP(1024) | 0.8473 | 0.8397 | 2.2237 | 0.9583 | 12.0849 | 0.0388 |
| | FedCRAP(2048) | 0.8511 | 0.8429 | 2.7982 | 0.6400 | 11.0570 | 0.0247 |
| | FedCRAP(4096) | 0.7996 | 0.7997 | 4.2499 | 0.1176 | 9.2394 | 0.0110 |

The parenthesized numbers indicate the perturbation radius (in units of 1/255) prior to norm-clipping. For example, FedEM(1024) denotes a perturbation radius of 1024/255 for FedEM.

## A.4 DIFFERENT PERTURBATION RADIUS

To investigate how the perturbation radius — a core hyperparameter directly determining perturbation intensity and data distortion — influences the privacy-utility trade-off of FedCRAP and FedEM on medical datasets, we conducted experiments with radii 1024, 2048, and 4096 (unit: 1/255)

As shown in Table 3, We found that FedCRAP is more sensitive to radius changes and small perturbation radii (e.g., 1024) enable FedCRAP to strike a favorable privacy-utility trade-off: utility remains close to FedEM while privacy is significantly strengthened.

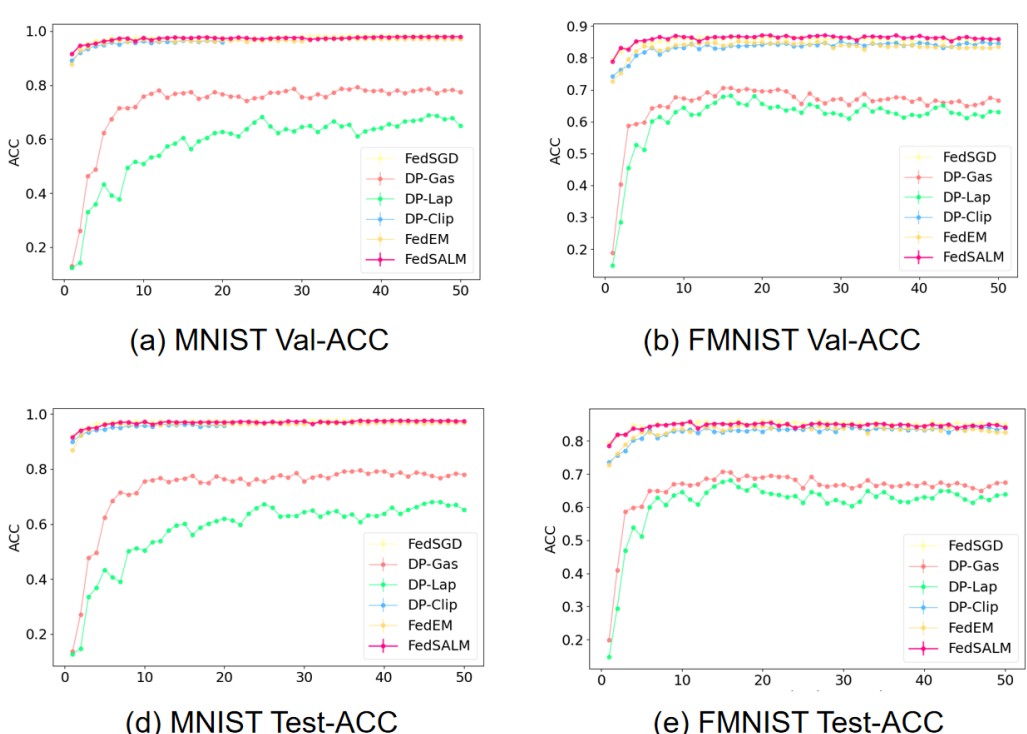

Figure 5: Accuracy Comparison of Privacy-Preserving Techniques.

