# OpenReview forum: "FedCRAP: Federated Critical-Region-Aware Perturbations for Refined Privacy-Preserving Federated Learning"
_ICLR.cc/2026/Conference — Submitted to ICLR 2026_

### Official Review · Reviewer_8jqa · 2025-10-22

**Soundness:** 2
**Presentation:** 2
**Contribution:** 2
**Rating:** 6
**Confidence:** 3

**Summary:**

This paper proposes FedCRAP, a novel privacy-preserving technique for Federated Learning (FL) designed to defend against gradient leakage attacks (GLAs). The core idea is to move away from adding uniform noise (as in standard DP) and instead apply region-aware perturbations. The method identifies the "critical" regions in a user's data by analyzing gradient magnitudes and then injects noise only into these salient regions.. This approach is motivated by its potential for sparse data, such as medical images, where critical features are often localized. The goal is to maximize privacy (reconstruction hardness) while minimizing the loss in model accuracy.

**Strengths:**

1. **Novel approach:** Introduces an innovative region-aware perturbation strategy that goes beyond traditional global noise injection in differential privacy. The gradient-guided sparsity pattern identification is conceptually sound and effective.

2. **Clear motivation and intuition:** Provides a strong rationale — showing that uniform noise is inefficient in low-information regions and under-protective in sensitive areas. The proposed gradient-guided, targeted noise injection offers an intuitive and well-justified solution.

3. **Relevant problem domain:** Addresses a critical and practical challenge in FL, focusing on protecting sensitive medical data where privacy is essential.

4. **Positive empirical results (vs. DLG):** Demonstrates improved privacy preservation (higher MSE, lower SSIM) against DLG attacks on MNIST, FashionMNIST, and MedMNIST datasets, while maintaining model accuracy comparable to standard or unprotected methods.

**Weaknesses:**

1. **Lack of formal privacy guarantees:** The paper provides no rigorous theoretical privacy analysis.

2. **Limited experimental setup:** The evaluation does not reflect realistic FL scenarios, as the experiments are conducted on a very small scale using an IID data distribution.

3. **Narrow threat model:** Focuses mainly on DLG attacks with no adaptive-attack evaluation; the novelty over prior data-space perturbation methods may be incremental.

**Questions:**

1. **Formal privacy guarantees:** Can you provide formal privacy guarantees for FedCRAP? How does it compare to differential privacy in terms of theoretical foundations and provable privacy bounds?

2. **Adaptive attack robustness:** How does the method perform against adaptive attacks that are aware of the region-aware perturbation strategy? Has its robustness been evaluated under such informed adversarial conditions?

3. **Performance on real clinical data:** It would be interesting to know how FedCRAP performs on real-world clinical datasets, beyond benchmark datasets like MedMNIST. Does it maintain both privacy protection and model utility in complex, heterogeneous medical data scenarios?

---

### Official Review · Reviewer_3EqM · 2025-10-22

**Soundness:** 2
**Presentation:** 1
**Contribution:** 2
**Rating:** 2
**Confidence:** 5

**Summary:**

This paper proposes Federated Critical-Region-Aware Perturbations (FedCRAP), a defense mechanism for Federated Learning (FL) against gradient inversion attacks, particularly focusing on the limitations of global noise injection methods, which can harm utility or under-protect critical areas, especially in sparse medical data. FedCRAP introduces a new approach that strategically injects noise only into task-critical image regions identified by high gradient magnitudes, aiming to align perturbations with data sparsity and provide targeted protection. By concentrating perturbations on sensitive areas using a dynamically generated mask based on gradient thresholds, the method claims to achieve a more favorable balance between privacy preservation against attacks like DLG and maintaining model utility. Experiments on datasets including MNIST, FashionMNIST, and medical imaging datasets reportedly demonstrate FedCRAP's ability to preserve model accuracy while significantly reducing privacy leakage risks compared to state-of-the-art methods.

**Strengths:**

* This paper identifies a limitation in existing FL privacy methods, which often apply uniform noise globally, potentially harming utility or under-protecting critical regions, especially in sparse data like medical images. FedCRAP proposes a targeted perturbation strategy to address this specific issue.

* The core idea of aligning perturbations with task-critical regions, identified via gradient magnitudes, is intuitively appealing. This approach aims to provide stronger protection where it is most needed while potentially preserving utility in less critical areas, offering a more refined balance than the global noise injection.

**Weaknesses:**

* Unprofessional Acronym and Presentation: The chosen acronym "FedCRAP" is highly unprofessional and detracts significantly from the paper's credibility. Additionally, figures like Figure 1 appear rudimentary and lack the polish expected for a top-tier conference submission.

* The core technical contribution appears weak. The paper explicitly states its region-aware insight builds upon Sun et al. (2024b) and uses the method from that same work for mask generation. FedCRAP seems primarily focused on adapting an existing localized perturbation/masking technique (originally for unlearnable examples) to the FL privacy setting, rather than proposing a fundamentally new defense mechanism. The listed contributions also seem padded, with "Problem Identification" and "Empirical Validation" not representing core technical innovations.

* Section 3.1, which defines standard Federated Learning , is inappropriately placed within the methodology section instead of in the background or preliminaries. The description of the actual FedCRAP algorithm (Section 3.3) is quite brief, lacking sufficient detail on the integration of the bi-level optimization and the practical implementation of the dynamic masking and perturbation updates.

* The paper completely lacks essential theoretical analyses. There is no formal security analysis or proof regarding the privacy guarantees offered against gradient inversion attacks (e.g., in terms of differential privacy bounds or other metrics). Furthermore, there is no analysis of the computational complexity, communication overhead, or convergence properties of the proposed method, despite claiming the overhead is "minor".

* While the paper compares against DP variants and FedEM, the evaluation is limited. The datasets used (MNIST, FashionMNIST, some MedMNIST variants)  are relatively simple image classification tasks. The experiments are conducted under a basic IID data split across only 4 clients, which does not reflect the heterogeneity and scale challenges common in real-world FL.

* The abstract claims "clear superiority over previous state-of-the-art (SoTA) methods". However, the results in Table 1 and Table 2  show a more nuanced picture. While FedCRAP often performs well on privacy metrics (like Feature MSE), it doesn't uniformly dominate all baselines across all metrics and datasets (e.g., FedEM sometimes shows better utility, and standard DP sometimes achieves higher privacy scores on certain metrics). The claim of "clear superiority" seems exaggerated based on the presented data.

* The paper suffers from numerous grammatical errors and inconsistent formatting. Examples include missing spaces before parentheses, inconsistent capitalization, and awkward phrasing, which detract from the paper's professionalism and readability.

**Questions:**

* Given that the core region-aware masking mechanism is adapted from Sun et al. (2024b) , could you elaborate on the primary technical novelty of FedCRAP beyond this adaptation to the FL privacy setting?

* Why does the paper lack a formal analysis providing rigorous privacy guarantees (e.g., in terms of differential privacy) or analyzing the computational complexity and convergence behavior of FedCRAP?

* How would the results obtained on relatively simple datasets under IID conditions with only four clients generalize to the more complex, heterogeneous, and large-scale scenarios typical of real-world federated learning?

* Considering the results where FedCRAP does not uniformly outperform all baselines across every metric, could you please provide a stronger justification for the claim of "clear superiority"?

* What was the rationale for placing the standard definition of Federated Learning within the methodology section (Section 3.1)  and keeping the core algorithm description relatively brief?

---

### Official Review · Reviewer_WrB4 · 2025-10-28

**Soundness:** 2
**Presentation:** 2
**Contribution:** 2
**Rating:** 2
**Confidence:** 4

**Summary:**

This paper investigates defenses against gradient inversion attacks in federated learning by introducing perturbations. It proposes a novel framework that leverages gradient-guided sparsity patterns. The goal is to strike a balance between privacy preservation and model utility. The proposed method is evaluated on four public datasets and shows improved performance over baseline approaches.

**Strengths:**

- Privacy protection in federated learning is a critical and challenging research area.
- The idea of using gradient information to guide the defense mechanism is conceptually clear and intuitive.
- The paper provides a a comprehensive review of related work, helping readers understand the context and motivation.

**Weaknesses:**

- It would be beneficial to include an algorithm box to clearly present the overall workflow and key steps of the proposed method.

- The perturbation update and mask generation appear to rely heavily on existing techniques. The specific technical contributions of this work should be more clearly distinguished and emphasized.

- The method requires fine-grained generation of perturbations, which may incur computational overhead. A detailed analysis of computational cost and latency on local clients is necessary. Does this approach introduce significant burden in practical deployments?

- The evaluation only considers DLG as the attack method. However, DLG may not represent the strongest or most up-to-date attack. The paper should include comparisons with more recent and powerful attack methods to validate the robustness of the proposed defense.

- The experimental evaluation is relatively weak. It mostly focuses on performance comparison across methods. Additional experiments would strengthen the empirical evidence.

- The datasets used are mainly from MedMNIST, which consists of low-resolution medical images. However, real-world medical images are typically high-resolution. The practical applicability of the proposed method to real-world medical scenarios requires more rigorous evaluation.

**Questions:**

-	DP-based methods offer mathematically provable privacy guarantees. It is unclear how this method compares in terms of privacy rigor. Can any theoretical privacy guarantees or bounds be provided?

---

### Official Review · Reviewer_meXv · 2025-10-31

**Soundness:** 2
**Presentation:** 1
**Contribution:** 2
**Rating:** 2
**Confidence:** 4

**Summary:**

The paper proposes FedCRAP, a framework for federated learning that selectively injects perturbations into task-critical image regions identified via gradient magnitudes. FedCRAP introduces gradient-guided masking, localized PGD-based noise generation, and sparsity constraints to protect privacy against Gradient Leakage Attacks (GLAs). Authors conduct experiments on MNIST-series to demonstrate FedCRAP maintains high utility while improving resistance to gradient inversion.

**Strengths:**

**S1.** The challenge of uniform global noise addition fails to consider spatial heterogeneity of medical image is make sense.

**S2.** Experiments include comparisons against standard DP baselines and FedEM across multiple datasets.

**Weaknesses:**

**W1.** The quality and clarity of Figure 1 are too low. Authors are encouraged to redraw it.

**W2.** The proposed algorithm ***(gradient-magnitude masking with localized PGD noise)*** is an incremental variant of prior gradient-guided adversarial masking and region-aware perturbation methods. In addition, there is no proof or quantifiable privacy guaratee.

**W3.** (1) The selection of the gradient threshold $\tau$ and perturbation radius $\rho$ is ad hoc, with tuning results shown only for FashionMNIST, (2) There is no clear sensitivity analysis on why $\tau = 10\%$ is best across datasets, and (3) The dependence of privacy–utility balance on these parameters is not clear.

**W4.** Datasets like MNIST and FashionMNIST still overly simplistic, and no experiments on non-iid splits that are critical for real-world FL. Furthermore, the evaluation of true attach resilience is missing.

**W5.** Without pseudocode or reproducible settings.

**Questions:**

Please see Weaknesses.

Additional questions:

**Q1.** Can the authors formally analyze the privacy guarantee of FedCRAP?

**Q2.** How does the algorithm behave under non-IID client distributions? Does region-level sensitivity generalize when data vary across clients?

---

### Meta-Review · Area_Chair_jZtv · 2026-01-05

**Summary:**

The paper proposes a federated learning framework that selectively injects perturbations into specific image regions identified via gradient magnitudes for specific tasks. The reviewers raise the concerns in paper writing/organization, incremental technical novelty, lack of formal privacy guarantees and theoretical analysis, limited experimental validation, etc. As there is no rebuttal response and discussion, this would be a clear rejection.

**Reviewer Concerns:**

There is no rebuttal response provided and no discussion either.

**Reviewer Scores:**

There is no rebuttal response provided and no discussion either.

---

### Decision · Program_Chairs · 2026-01-26

Reject